# Dynamic Properties of the DNA Damage Response Mre11/Rad50 Complex

**DOI:** 10.3390/ijms241512377

**Published:** 2023-08-03

**Authors:** Jacopo Vertemara, Renata Tisi

**Affiliations:** Department of Biotechnology and Biosciences, University of Milano-Bicocca, 20126 Milan, Italy; jacopo.vertemara@unimib.it

**Keywords:** resection, DNA tethering, MRX, MRN, ATM, Tel1, DDR, DSBs

## Abstract

DNA double-strand breaks (DSBs) are a significant threat to cell viability due to the induction of genome instability and the potential loss of genetic information. One of the key players for early DNA damage response is the conserved Mre11/Rad50 Nbs1/Xrs2 (MRN/X) complex, which is quickly recruited to the DNA’s ruptured ends and is required for their tethering and their subsequent repair via different pathways. The MRN/X complex associates with several other proteins to exert its functions, but it also exploits sophisticated internal dynamic properties to orchestrate the several steps required to address the damage. In this review, we summarize the intrinsic molecular features of the MRN/X complex through biophysical, structural, and computational analyses in order to describe the conformational transitions that allow for this complex to accomplish its multiple functions.

## 1. Introduction

Accidental or programmed DNA double-strand breaks (DSBs) expose the cell to the danger of losing genetic information or to impromptu chromosome rearrangements. Both prokaryotic and eukaryotic cells can undergo DSB repair via different mechanisms [1,2], sharing some of the conserved first responders to the damage such as Ku proteins and Mre11-Rad50-NBS1 (in mammals)/Xrs2 (in yeast) (MRN/X) complexes [3]. Ku proteins protect the DNA ends from degradation and trigger non-homologous end-joining (NHEJ) repair [4]. In contrast, the MRN/X complex is involved in the removal of Ku proteins from the DNA ends and in the resection of the protein-blocked DNA ends, i.e., the generation of a 3′ single-stranded DNA end that is required for Homologous Recombination (HR) [1] (Figure 1). The MRN/X complex is also involved in the resolution of several other aberrant structures that can arise due to DNA replication stress or DNA breaks generated during meiosis (for a review, see [5]).

The MRN/X complex is highly conserved and recruits several proteins that are required to activate the DNA damage checkpoint, such as the ATM/Tel1 kinase, or long-range DNA resection (up to more than 1000 nucleotides), such as Exo1 or Dna2 exonucleases [6]. The complex is a heterohexamer composed of two homodimers of Rad50 and Mre11 subunits, bound to one or two Nbs1/Xrs2 subunits that are only present in eukaryotes [7]. While the Rad50 dimer is competent for DNA scanning and DNA tethering functions [8], Mre11 is required to recognize the DNA end and to exert either the exo- or endonuclease activities (requiring association with Sae2/CtIP) involved in the short-range resection that releases the DNA end from any protein or chemical adducts [9]. MRN/X can activate the DNA damage checkpoint kinase Tel1/ATM when bound to naked DNA, but not to nucleosome-bound DNA, although MRN/X is actually able to bind histone-associated DNA [10] and to resect DSBs in heterochromatin regions as well [11]. The mechanism of ATM/Tel1 activation is unfortunately still unclear, but it involves recruitment to the DNA by MRN/X [10,12,13,14,15].

Tel1/ATM and its cognate kinase Mec1/ATR are members of the phosphatidylinositol-3-kinase-related protein kinases (PIKKs) family. They master the DNA damage checkpoint response, controlling resection extension via the phosphorylation of Rad9 (53BP1 in human) and Rad53 (CHK1 in human), and leading to cell cycle arrest in case the DNA DSB is not repaired (for a review on Tel1/ATM signaling, see [16]). MRN/X and Tel1/ATM are also necessary for maintaining the telomere length [17].

The MRN/X complex undergoes dramatic conformational rearrangements to allow for physical contact between its subunits and different interactors, both proteins and DNA. In this review, we summarize the current models of the several states that the Mre11/Rad50 complex adopts when responding to a DNA DSB in order to orchestrate the DNA damage response and allow for DSB repair, and the issues that are still under debate arising by the latest findings on the MRN/X structure and function.

## 2. Rad50 Intrinsic Dynamic Properties

Rad50 belongs to the ABC-ATPase protein family, which displays an N-terminal Walker A motif and a C-terminal Walker B motif that concur in forming the Nucleotide Binding Domain (NBD). In particular, each Rad50 subunit in a dimer participates in constituting the binding sites for two molecules of ATP, which are part of the dimerization interface (Figure 2A). In particular, residues in the conserved motifs, the Walker A and the signature, of opposite Rad50 subunits make several contacts with the ATP molecule (Figure 2B), which is a significant player in the dimeric interface. The network of contacts through the same subunit and between the two opposite subunits establishes a tight cooperativity between the two active sites [18]. The N-terminal and C-terminal lobes can rotate on each other to adopt two different conformations, depending on ATP binding/hydrolysis, thanks to the torsion of the hinge helix that ensures communication between Walker A, Walker B, and the signature motif S(G/A)G [19]. The ATP-bound state is the most prone to dimerization, while ATP hydrolysis affects both the interface involving the two ATP molecules and the conformation of the globular domain of Rad50, lowering the affinity of a Rad50 subunit for the other [20,21,22,23].

The central region of Rad50 constitutes a 20 nm (in bacteria) to 60 nm (in humans) long coiled-coil rod that bends on itself at the zinc hook, where two conserved residues of Cysteine in a conserved CPXC motif (where X is a hydrophobic residue) are involved in coordinating a Zn^2+^ ion.

### 2.1. A Dynamic Allosteric Pattern Is the Key to Rad50 Conformational Change

ATP binding to Rad50 induces a conformational change that dramatically increases the Rad50 NBD affinity for itself, allowing for its dimerization [24,25,26]. The analyses via nuclear magnetic resonance (NMR) spectroscopy of mutants in the hinge region and the basic switch conserved Arg805 residue of *Pyrococcus furiosus* Rad50 (corresponding to *Saccharomyces cerevisiae* Arg1217 and human Arg1214) revealed an allosteric network of clustering residues in the α1-β4 loop within the core of the Rad50 globular domain and facing the ATP binding site and the base of the coiled coil. This network of residues influences conformational changes in Rad50 that are related to its dimerization and ATP hydrolysis activity, finally leading to the activation of the Mre11 exonuclease [21,27].

Consistently, with these observations, molecular dynamics simulations revealed that the α1-β4 loop shows higher mobility in *S. cerevisiae* Mre11/Rad50-ADP than in Mre11/Rad50-ATP. A mutation that affects the flexibility of the α1-β4 loop, i.e., the A78T mutation in *S. cerevisiae* Rad50, disrupts the Rad50 core allosteric network, allowing for Rad50 to undergo the conformational change even in the presence of ATP, and causing the drift of the Mre11 dimer from the Rad50 globular domains [28].

**Figure 2 ijms-24-12377-f002:**
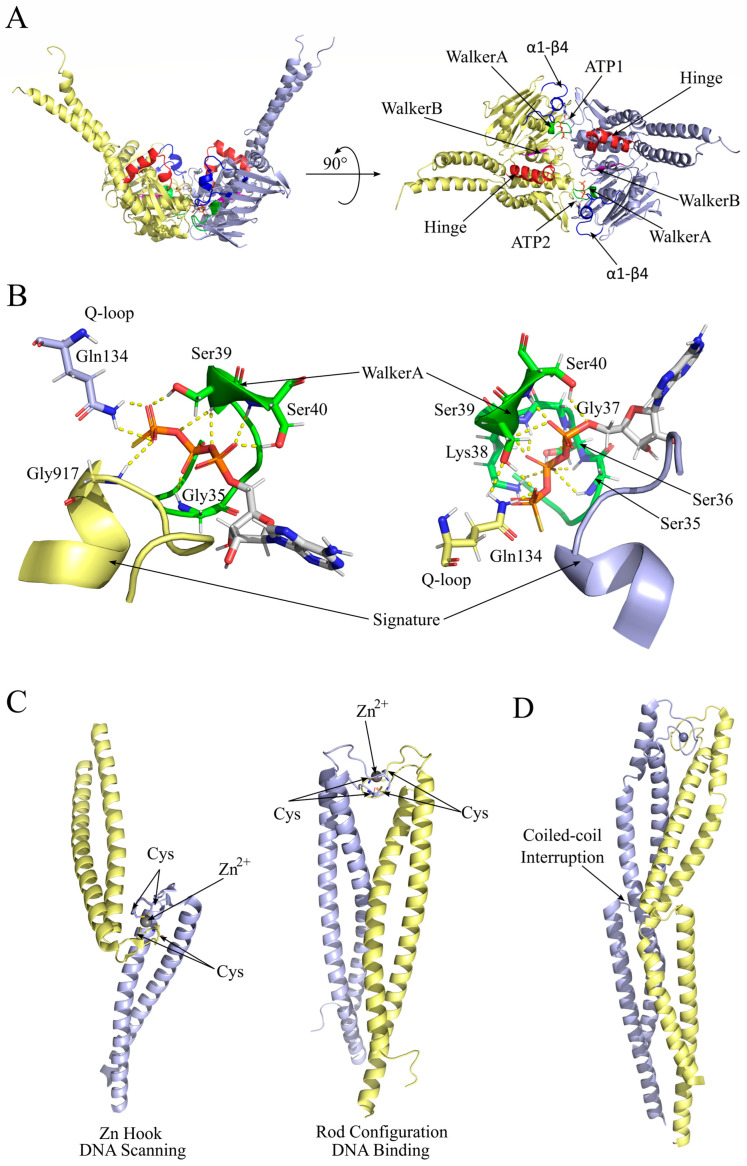
Rad50 dimerization requires ATP binding and conformational rearrangements. (**A**) Frontal (**left**) and top (**right**) view of a cartoon representation of *Methanocaldococcus jannaschii* Rad50 dimer (PDB ID: 5DNY) encompassing two molecules of ATP at the interface (represented as sticks with carbon in grey, oxygen in red, nitrogen in blue, and phosphorus in orange). The two Rad50 subunits are represented as yellow and violet cartoons. Walker A is in green, Walker B is in magenta, the α1-β4 loop is in blue, the Q-loop is in cyan, and the hinge helix is in red. The distal portion of the coiled-coil region was not resolved. (**B**) Detail of the dimeric interface comprising the ATP binding sites. Hydrogen bonds of the ATP molecule with the protein residues, in sticks, are represented as dashed yellow lines. The signature residues, conserved among the ABC ATPase family, are represented as cartoons. (**C**) Zn hook in Rad50 dimer can adopt different conformations that have been proposed to be involved in the transition of the *P. furiosus* coiled-coil ring, from a DNA-scanning oval-shaped ((**left**) PDB ID: 1L8D) conformation to a DNA-binding rod ((**right**) PDB ID: 6ZFF) conformation. (**D**) Human Rad50 Zn hook region showing the first interruption of the coiled coil at 45 aa from the Zn hook (PDB ID: 5GOX) [29].

### 2.2. Rad50 Coiled-Coil Region Has Roles in DNA Tethering

Most of the specific functions pertaining to the Mre11/Rad50 complex rely on specific features of the Rad50 globular domain. However, hypomorphic mutations were isolated in the coiled-coil hydrophobic patch surrounding the conserved cysteine residues in the Zn hook that affected functions that are typically associated with the globular domain, such as Tel1 (ATM) activation, DNA tethering, and DNA double-strand break end resection, suggesting additional roles for this region [30]. A similar phenotype was also observed in a mutant with a truncation of the coiled-coil domain but still allowing for the dimerization of the hook domain [31]. These mutants were particularly defective in intra-complex dimerization, which thus seems to be specifically required for Tel1/ATM activation. Since the extension of the coiled-coil arms has progressively increased in evolution, from 18 nm in phages to 60 nm in eukaryotes, it is possible that additional activities were acquired in these last organisms.

The Zn hook is necessary for the association of two coiled coils by sharing the metal ion among the four Cysteine residues of two Rad50 subunits belonging to the same complex (intra-complex bridge) [32], and it was proposed that it could be shared by two different complexes (inter-complex bridge) [33,34]. The Zn hook association is stabilized by the interactions through the distal coiled-coil rods [29] and is probably regulated via phosphorylation on Thr690 [35]. The transition from an oval-shaped ring to a rod was also observed when the complex bound to the DNA end; thus, the oval-shaped ring was proposed to be involved in genomic DNA scanning [8] (Figure 2C). The human RAD50 coiled coil is characterized by two interruptions, which confers an increased flexibility to the whole structure, allowing for the rapid and even sharp bending observed via atomic force microscopy (AFM) [32,36] (Figure 2D). Interestingly, a mutant affected by the deletion of both coiled-coil arms but still retaining the zinc hook region was unable to create DNA loops or bridges between two DNA ends, suggesting that the extension or the flexibility of the coiled-coil regions is also involved in establishing these contacts [37].

## 3. Mre11 Intrinsic Dynamic Properties

The Mre11 N-terminal nuclease domain contains five conserved loops displaying histidine and glutamate residues involved in the coordination of two Mn^2+^ ions required for its endonuclease and 3′–5′ exonuclease functions (Figure 3A) [38,39]. The central region of the Mre11 sequence contains the capping domain, which is involved in making contact with the 3′ DNA end. The Rad50 Binding Domain (RBD) is connected to the capping domain by a flexible and mainly unstructured linker and binds to the coiled coil in Rad50, ensuring the persistency of the Mre11/Rad50 complex after ATP hydrolysis [26] (see the following section for a description of this interaction).

In the molecular dynamics simulations, the capping domain rotation movement was observed in an Mre11 dimer (Figure 3B). The *S. cerevisiae* Mre11 R10T mutant, affecting the nuclease domain, showed altered dynamics in the capping domain in the MD simulations [40], revealing a complex dynamic relationship between the nuclease domain and the capping domain. A similar relationship was characterized by the NMR chemical shift analyses in mutants affected by a single amino acid substitution corresponding to a missense mutation found in cancer in human Mre11 [41,42]. The movement of the capping domain can be connected to the ability of Mre11/Rad50 complexes to facilitate the unwinding of the DNA end [43]. In fact, its hyper rotation in the Mre11 R10T mutant results in *sae2*Δ DNA damage sensitivity and resection defect suppression thanks to enhanced Exo1 exonuclease attack [40].

## 4. Mre11/Rad50 Complex Adopts Different Conformations for Different Functions

### 4.1. The Resting State Is Competent for DNA Ends Tethering, DNA Scanning, and Multicomplex Assembly

Once assembled through ATP binding, the Rad50 dimer offers a channel between its two coiled-coil rods to host a DNA molecule and can bind an Mre11 dimer on its other side, far away from the DNA molecule, making contact with the globular, the capping, and the RBD domains [24,44,45] (Figure 4A). It was immediately evident that in this configuration, the Mre11 dimer cannot access the DNA, since its nuclease sites are not only hindered by the Rad50 dimer, but are also directly involved in the Mre11/Rad50 protein–protein interaction [46] (Figure 4A).

The resting state was reported to be required both for DNA tethering and for Tel1/ATM activation [19]. Although initially, an inter-complex dimerization via the Rad50 Zn hooks was proposed to mediate the DNA bridging, the Zn hook was found to be a very stable inter-complex dimerization interface, and the apex Zn hook region of the coiled coil was recently proposed to mediate these intra-complex dimerizations [34].

Performing luminescence resonance energy transfer (LRET) experiments allowed us to obtain distance restraints that drove the computational reconstruction of three different states of MRad50 complexes in a dynamic equilibrium that is independent from the nature of the nucleotide and is observed in the presence of DNA as well [27,47]. It is unclear if the so-called ‘partially open’ state is only a transition state or if it is competent for specific functions exerted by the complex.

### 4.2. Cutting State Is Competent for DNA End Recognition and Nuclease Activity

The conformations of the Mre11/Rad50 complex that are competent for nuclease activities are the most recently resolved. First, a novel conformation defined as a ‘cutting state’ was resolved on a naked DNA end [22] (Figure 4B) and then, very recently, an analog conformation was resolved that could guarantee access to the DNA for the Mre11 nuclease sites that are also far from the DNA end by clamping the DNA molecule between the Rad50 dimer and the Mre11 dimer [23] (Figure 4C,D).

In both the DNA end-associated exonuclease-competent complex (Figure 4B) and the distal DNA-associated endonuclease-competent complex (Figure 4C,D), the DNA is bent to adapt to the complex channel and to approach the nuclease site. The Rad50 Binding Domain (RBD) is not the only region of Mre11 that still makes contact with Rad50, since, at least in bacterial complexes, a novel interface was described between the fastener loop in bacterial Mre11 and the external β-sheet of Rad50 [22] (Figure 4B). Unfortunately, this loop (aa 137–149 in *E. coli*) is not conserved in eukaryotes, and it remains to be established which contacts can stabilize the cutting state in these organisms.

### 4.3. Regulation of Mre11/Rad50 Complex Functions via Interaction with Different Partners

In eukaryotes, a plethora of interactors have developed to assist the Mre11/Rad50 functions in DNA damage as well as telomere maintenance.

The Xrs2/Nbs1 subunit is a constitutive partner in the eukaryotic MRN/X complex and acts as a largely disordered scaffold that recruits the various components of DNA repair foci [48,49] and stabilizes the Mre11 dimer by binding asymmetrically to the complex with M_2_R_2_X_1_ stoichiometry [34]. In *S. cerevisiae*, Xrs2 is largely dispensable but is required for Mre11 to enter the nucleus and for proper Tel1 signaling [50].

Many efforts were directed at clarifying the role of Sae2/CtIP, another largely disordered protein that exerts different functions alongside MRN/X complexes. Sae2 was not only reported to be required for MRN/X endonuclease activity [51,52], but also to contribute to DNA end tethering [53,54], to Dna2 exonuclease recruitment to the DSB [55], and to negatively regulate the DNA damage response by hindering Rad9 and Rad53 interaction [56]. Rad50 residues involved in the interaction with Sae2 were identified [57] and are in the neighborhood of the interface involved in the interaction with the Mre11 fastener loop in the *E. coli* Mre11/Rad50 complex homolog cutting state [22], suggesting that Sae2 can be a part of this interface in eukaryotes.

Recently, the Rad50 residue K81, involved in Sae2 interaction in yeast, was proven to be required for the interaction with another protein, Rif2. Rif2 is a member of the *S. cerevisiae* shelterin complex, whose function is to protect the telomere from DNA degradation by regulating the MRX complex [58,59]. Rif2 was found at the DNA DSB as well [60], where it was demonstrated to compete with Sae2 for MRX binding [61]. In fact, whereas Sae2 would be required to stabilize the cutting state, Rif2 binding seems to be incompatible with the transition into the cutting state [62,63]. The Rif2 partner at telomeres, Rap1, is also present at the DNA DSB and can influence MRX activity independently from Rif2 by a mechanism that is still uncharacterized [64]. Due to the different relative abundances of these two alternative partners at DSB DNA damage sites and at the telomeres, they would be good candidates to direct the distinct behavior of MRX complexes in these genomic loci.

Upon DNA DSB appearance in eukaryotic cells, MRN/X complexes rapidly assemble at the DNA end, forming foci together with their interactors, which are essential for DNA damage repair and checkpoint activation [65,66]. The mechanism driving this rapid relocalization of Mre11/Rad50 complexes is not clear yet. Recently, the MRN complex interacting protein (MRNIP) was shown to form liquid-like condensates that compartmentalize and concentrate the MRN complex in the nucleus and rapidly relocalize to the DSB facilitating the formation of foci [67].

### 4.4. The Topology of Mre11/Rad50 DNA Binding

Although it is clear that the globular domains of MRN/X complexes have the capacity to bind DNA, it is not definitely clear which part of them actually makes contact with the DNA end or the DNA in general. Since Rad50 is homologous to the SMC (Structural Maintenance of Chromosomes) proteins, which are involved in the regulation of chromosome segregation and chromatin organization via DNA bridging [68], it was tempting to assume that the DNA binding topology would be similar to what was observed in this class of proteins [24,45]. However, it is not clear if the MRN/X complex ring ever encloses DNA either as a single molecule or as a DNA loop extrusion (Figure 5A,B). Additionally, it was assessed that Mre11 is required for the recognition of the DNA end [69], but the mechanism of this recognition is not clear yet.

The recent characterization of *C. thermophilum* Mre11/Rad50 bound to Nbs1 revealed precious insights into the relationship between ATP and DNA binding. In detail, DNA binding was reported as ATP-independent for human MRN, but not for bacterial homologs. For the thermophilic filamentous fungus *C. thermophilum* Mre11/Rad50 (CtMR), DNA binding was proposed to involve two different binding motifs, one which preferentially binds the DNA ends in an ATP-dependent manner, and the other also involving Nbs1 and the Mre11 uncharacterized C-terminal region, which is independent of ATP. In any case, the complex fully engages ATP with the signature motif only upon DNA binding, resulting in the opening of the coiled-coil arms from a rod to a ring conformation and up to a 10-fold stimulation of ATP hydrolysis [34]. This is in agreement with the previously reported structure for the ATP-bound state of archaean MRN, where the coiled-coil domains are also open [45]. It is yet to be established if ATP hydrolysis would be sufficient to trigger the conformation switch from the resting state to the cutting state, and what kind of contact with DNA would be necessary to elicit ATP hydrolysis, with intact DNA or with the DNA ends. Additionally, the cutting state showed a rod conformation of the coiled-coil arms [22], suggesting that after scanning the DNA, the complex stops, possibly by recognizing the DNA end itself or by clashing with some other protein that is stuck to it (e.g., Ku, Spo11, or Sae2 itself), and clamp on the DNA.

*S. cerevisiae* MRX was found to oligomerize thanks to previously unrevealed Rad50 head intermolecular contacts, again facilitating foci formation in vivo [70]. No oligomerization was observed for MRN/X complexes involving the formation of larger rings, as was proposed for SMC1 and SMC3 [71]. Oligomerization was proposed to be the main state of Mre11/Rad50 (MR) complexes upon DNA damage, but DNA binding would shift them from multimers to (M_2_R_2_)_2_ dimers, which then reassemble along the DNA molecule; however, multimers were shown to be able to bind, tether, and displace DNA targets in real-time AFM [37,70].

A role was proposed for the zinc hook apex of the archaeal Rad50 coiled-coil region in contacting DNA and to not only allow for DNA binding, but also to allow for translocation of the oligomer along the DNA molecule, scanning for DNA DSBs [37] (Figure 5C), although no clear structural mechanism has been hitherto identified for this property. Molecular dynamics (MD) simulations seemed to confirm that the archaeal zinc hook region would be capable of tracking along dsDNA using its Rad50 zinc hook apex [37]. Additionally, the apex Zn hook was structurally characterized as mediating the tetramerization of CtMR complexes assembling in sheets along two tethered DNA molecules [34] (Figure 5D).

Mre11-Rad50 oligomerization was proposed as being able to allow for simultaneous endonucleolytic cleavage at different sites on the 5’ DNA strand near DSBs, since mutants that are defective in oligomerization do not possess this ability [70]. It is to be considered that oligomerization seems to involve the very same Rad50 β-sheet, which would be involved in Mre11 docking in the endonuclease’s activity-competent cutting state, leaving only one of the two Rad50 molecules free to make contact with another tetramer along the DNA molecule, which would disrupt the oligomeric state. This suggests that the oligomers would be prone to the delivery of the MRN/X complex at longer distances along the DNA molecule.

MRX complex oligomerization was also proposed to be involved in the amplification of the DNA damage checkpoint signal, since a mutant Rad50 protein that is defective in oligomerization is also defective in Tel1 activation [70]. It is not yet clear how the heterotetramer or the MRX oligomers would transduce the signal to the Tel1 protein, which is a huge protein as well (a dimer of two 2787 aa proteins in yeast) [72] (see Section 4.6).

### 4.5. The Dynamic Path from Resting State to Cutting State in DNA-Bound MR Complex Is Still under Debate

Questions arise when the topological path allowing for Mre11 to reach the DNA molecule away from the DNA end is considered (Figure 6).

In the cutting state, the Mre11 dimer as a solid is rotated with respect to the Rad50 dimer globular domains through a yet unclear path, since the DNA molecule is then situated between the Rad50 dimer and the Mre11 dimer. Only one of the two Mre11 subunits reaches the DNA, in an elegant demonstration that, although we instinctually appreciate symmetry, nature does not actually need it. As a matter of fact, during MD simulations, the two Mre11 subunits do not behave symmetrically, and one is always drifting first and further from the Rad50 counterparts [28]. This is consistent with the asymmetrical rotation of the Mre11 dimer with respect to the Rad50 dimer required to adopt the cutting state conformation.

Previous experimental analyses suggested that the Mre11 dimer is very stable [73], which would prevent the DNA passage between the two Mre11 subunits even if they could revolve around the Rad50 dimer (Figure 6A). As well, the Rad50/Mre11 interaction at the RBD interface is regarded as stable, since mutations impairing this interaction impinge on MRX activity [26,28]. On the contrary, the Rad50 dimer is known to be destabilized upon ATP hydrolysis, and the DNA passage between Rad50 subunits to reach the Mre11 catalytic site was previously proposed as a possible path [73] (Figure 6B). In order to assume the currently accepted final conformation of the cutting state, the Rad50 subunits would need to rotate on themselves, which could be impeded by the hindrance of the long coiled-coil arms. Both of these paths would indeed create the clamp on the DNA molecule, which would require the complex to either travel to the end of the DNA or to reverse the ‘clamping’ process in order to be released.

### 4.6. The Mechanism of Tel1/ATM Activation Is Still under Investigation

Tel1/ATM is recruited to the DNA through high affinity binding to the C-terminal of Nbs1/Xrs2 [13,14] and low-affinity interactions with the N-terminal forkhead-associated (FHA) domains of Nbs1/Xrs2 but also with Mre11 and Rad50 [10,15]. However, fusing the C-terminal Tel1-interacting domain from Xrs2 to a nuclear localization domain (NLS)-containing Mre11 is sufficient to restore normal Mre11/Rad50 complex association to DNA, telomere maintenance, and Tel1 signaling in yeast Xrs2-deficient cells. This suggests that the role of Xrs1, besides driving Mre11 into the nuclear compartment, is merely to enrich the Tel1 molecule presence near the MRX complex, which, in turn, stabilizes the presence of MRX on the DNA [74].

Although it is clear that Tel1/ATM activation required its recruitment to damaged DNA and MRN/X complexes [75,76], the necessity of Rad50-mediated ATP hydrolysis [10] and the unwinding of DNA ends [12] for Tel1/ATM stimulation were reported, although the resting state conformation seems to be required for Tel1 activation [19]. Consistently, Rad50 A78T mutation in *S. cerevisiae*, which destabilizes the resting state of the Mre11/Rad50 tetramer as described above, was also reported to counteract Tel1 activity [28]. Moreover, Rif2 activates Rad50 ATPase and competes for Tel1 binding to MRX [60,63], suggesting that both proteins contact MRX in the ATP-bound form.

Double-stranded DNA and MRX were described to activate Tel1 synergistically in yeast, although the full activation was obtained with long, nucleosome-free DNA. Tel1 activation did not require any dsDNA termini, but it required ATP hydrolysis via Rad50 [10]. These two requirements would ensure that Tel1/ATM activation does not occur along the intact chromosomal DNA, since nucleosome-free DNA and Rad50 ATPase activation are only available at the DNA DSBs [77] or at short telomeres [58].

## 5. Conclusions and Perspectives

While the role of MRN/X complexes in DNA DSBs repair has been studied for decades, the molecular details of MRN/X complex loading on DNA, chromatin scanning, and DNA end recognition are still under investigation. Foci formation and oligomerization add further complexity to the spatial segregation of free and DNA-bound MRN/X, raising more questions about the mechanism of DNA tethering and Tel1/ATM activation.

Further work is required to clarify the fate of the clusters of MRN/X complexes that accumulate at the DNA end. It is unclear if they all undergo ATP hydrolysis, if they are released from the DNA, or if they proceed with DNA scanning once the lesion is repaired. The recycling of MRN/X after hydrolysis is also unclear, since it is not established if Rad50 undergoes a spontaneous cycle of ADP/ATP exchange or if this process requires the intervention of some other proteins.

Furthermore, novel roles have recently emerged in different processes such as chromatin organization and transcriptional regulation [78]. Future research will have to address the implications of this novel role with respect to DNA damage checkpoint control and telomere maintenance, to enlighten how the MRN/X complexes can manage to discriminate between intergenic regions, DNA DSBs ends, and telomeres to perfectly tune the proper response.

The relevance of DNA damage and DNA repair topics is related to their correlation to cancer [79] and human genetic diseases such as ataxia telangiectasia-like disorder (ATLD), Nijmegen breakage syndrome (NBS), and NBS-like disorder (NBSLD). A fine understanding of the molecular mechanisms at the base of these phenomena is required in order to target their role in pathogenesis.

## Figures and Tables

**Figure 1 ijms-24-12377-f001:**
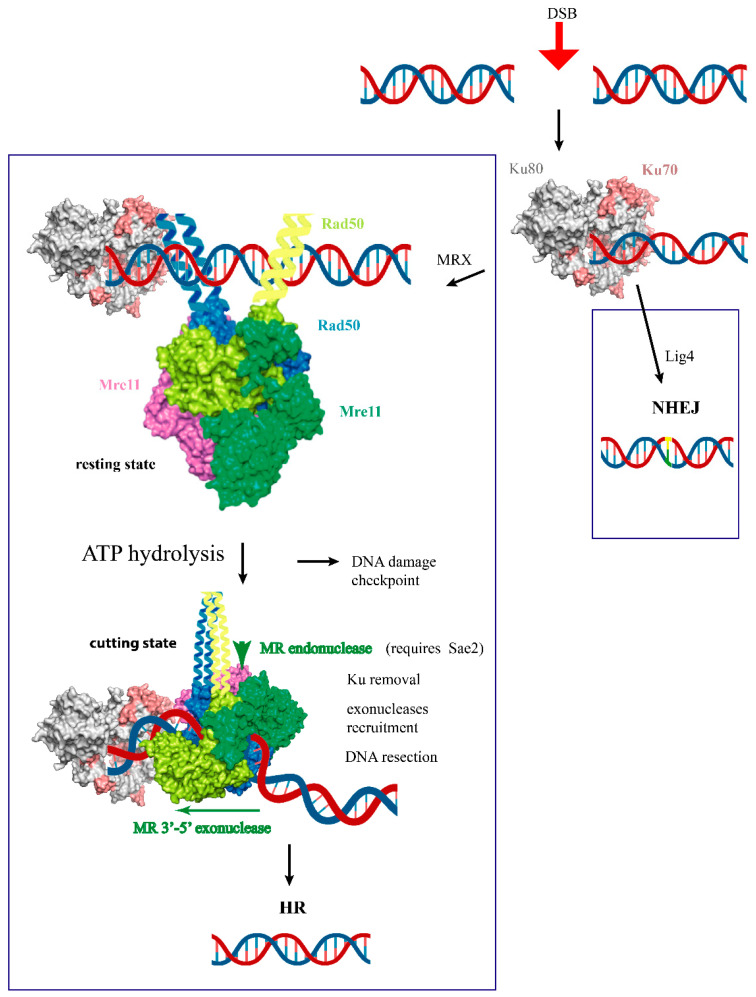
Early stages of DNA damage response in budding yeast model organism. Upon the insurgence of a DNA double-strand break (DSB), Ku proteins immediately bind to the free DNA ends and can contribute, at least in *Escherichia coli* and yeast, to the tethering of the DNA ends. Ku proteins are required for non-homologous end-joining (NHEJ) repair of the DNA DSB, catalyzed by a complex of proteins encompassing a DNA ligase IV (Lig4 in yeast) and DNA-PK in eukaryotes (although absent in yeast), which can lead to the introduction of errors in the DNA molecule. MRX binds to the DSBs in an ATP-dependent resting state that can tether the two DNA ends together. Upon Rad50-mediated ATP hydrolysis, MRX (only Mre11/Rad50 subunits are represented, with only proximal Rad50 coiled coil shown) adopts a cutting state that is competent for endonuclease activity, which also requires Sae2/CtIP and 3′–5′ exonuclease in order to release Ku-blocked DNA ends. Long-range resection requires the recruitment of different exonucleases (see text). DNA resection is necessary for the onset of the DNA DSB repair via Homologous Recombination (HR), which allows for the restoration of the undamaged DNA molecule. If DNA resection cannot proceed, MRX activates the DNA damage checkpoint that stalls the cell cycle.

**Figure 3 ijms-24-12377-f003:**
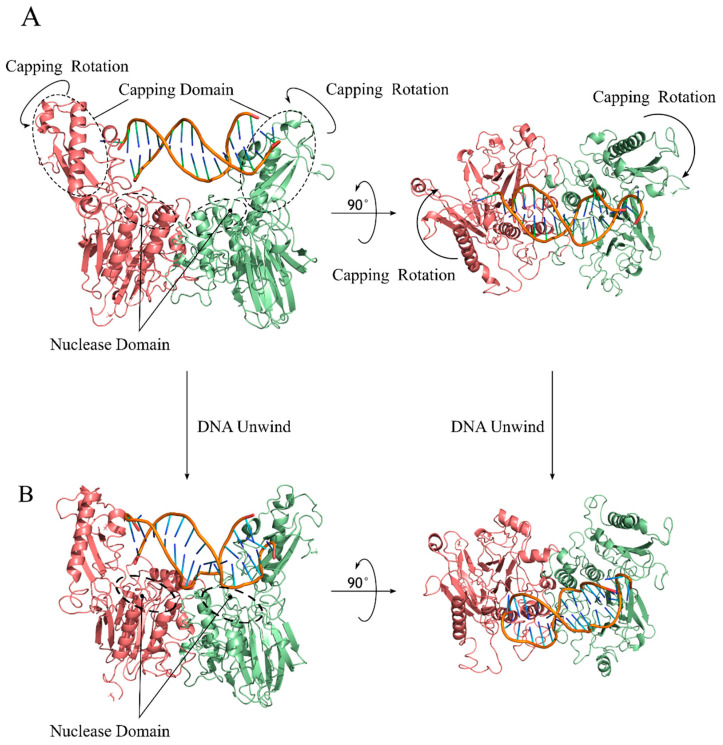
Mre11 capping domain rotation is able to unwind the DNA end. (**A**) Model of *S. cerevisiae* Mre11 dimer encompassing nuclease and capping domains, bound to a DNA molecule, showing the capping domain making contact with the DNA 3′ end. The capping domains were observed to rotate, as indicated by the arrows, during MD simulations [40]. (**B**) The final conformation of the Mre11 dimer bound to DNA revealed that the movement of the capping domain allows for Mre11 to unwind the DNA ends and to bend the DNA molecule while pushing the 3′ ended filament towards the catalytic site of Mre11 nuclease [40].

**Figure 4 ijms-24-12377-f004:**
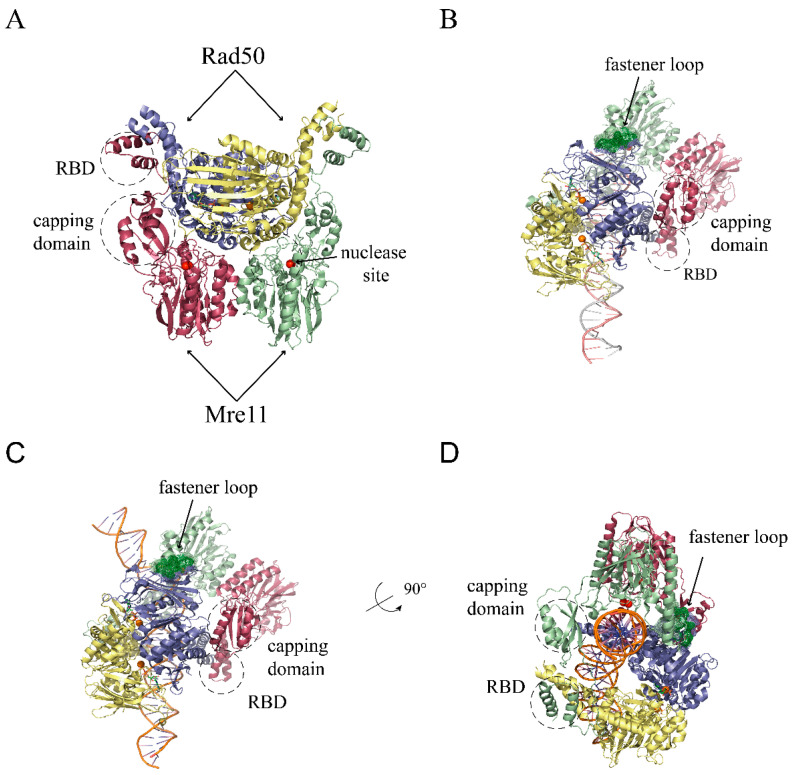
Mre11/Rad50 complex resting and cutting state structures as resolved via Cryo-EM. (**A**) Resting state of *E. coli* Mre11/Rad50 complex homolog (SbcCD) (PDB ID: 6S6V) [22]. Rad50 subunits are represented as yellow and violet cartoons, and Mre11 subunits are represented as red and green cartoons. The aa 140–147 and 152–153 of Mre11 subunits are not resolved in the structure. (**B**) Cutting state of *E. coli* Mre11/Rad50 complex homolog bound to a free end of a DNA molecule (PDB ID: 6S85) [22]. (**C**) Cutting state of *E. coli* Mre11/Rad50 complex homolog bound to a protein-blocked DNA molecule (PDB ID: 7Z03) [23]. (**D**) View of the same molecule as in panel C, rotated as indicated to visualize the active site of the Mre11 subunit bound to DNA. In all structures, only proximal regions of Rad50 coiled coils were resolved. In all panels, Mn^2+^ ions are represented as red spheres, Mg^2+^ ions are represented as orange spheres, and nucleotides are represented as sticks with carbon in green, oxygen in red, nitrogen in blue, and phosphorus in orange; the fastener loop in the Mre11 subunit docked to Rad50 is represented as a dark green mesh.

**Figure 5 ijms-24-12377-f005:**
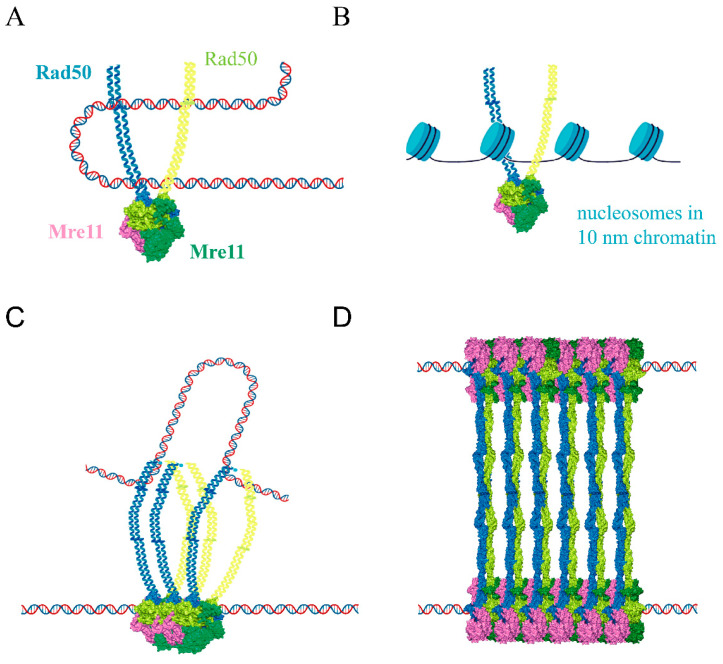
Possible topologies of MR complex resting state binding to a DNA molecule. (**A**) Rad50 oval ring coiled coil is large enough to host a naked DNA loop extrusion. (**B**) Eukaryotic Rad50 oval ring coiled coil is large enough to host a single 10 nm chromatin fiber. (**C**) Oligomeric MR complexes were observed to make contact with DNA both at the globular domains and at the apex of the coiled-coil region [37]. (**D**) DNA tethering was observed to be exerted by a ‘sheet’ of MR complexes bound to DNA and by apex zinc-hook-mediated tetramerization [34]. The cartoons of rod-shaped MR complexes are based on the structure of *Chaetomium thermophilum* Mre11/Rad50 complex (PDB ID: 7ZR1).

**Figure 6 ijms-24-12377-f006:**
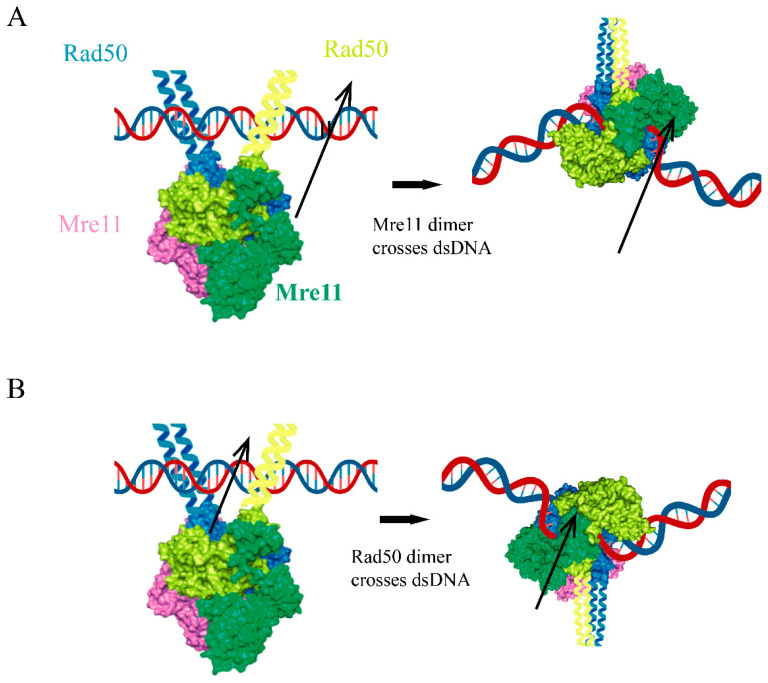
Models for MR complex resting to cutting state conversion when binding a DNA molecule far from the DNA end. (**A**) The Mre11 dimer dissolves, and the Mre11 subunits rotate over both the Rad50 dimer and the DNA molecule. (**B**) The Rad50 dimer dissolves upon ATP hydrolysis and the two Rad50 molecules move around the DNA molecule and then re-associate.

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
