# Peer review of "Dynamic Properties of the DNA Damage Response Mre11/Rad50 Complex"

_ijms, 2023, doi:10.3390/ijms241512377_

Round 1

Reviewer 1 Report

See attached.

Author Response

-This manuscript is globally pleasant to read, yet looks more focused on the MR part than on the overall MRN/X complex.

The reviewer understood correctly the focus of the review. The topic of DNA damage response is very broad, and we decided to narrow this review on the dynamics properties of the Mre11/Rad50 complex, that we have been studying for the last years.

-There are many points that should be clarified before publication:

- Along the manuscript, the authors refer to MN, MNR, and MNR/X complexes. It is hard to follow the flow of information when jumping from on to the other. The manuscript should be more organized to guide the reader through it.

We apologize for the inconsistency. MR abbreviation was limited to the necessary, and we wrote Mre11/Rad50 instead, to avoid confusion. MRX was used when strictly reporting data from budding yeast research, and this was clearly mentioned in the text. MRN was used when mammalian cells-based data were mentioned, that were not experimentally proven in yeast. This was explicated at the beginning of the introduction.

- line 29. “MRN/X complex is also involved in the resolution of several other aberrant structures that can arise during replication or meiosis”: could you give a couple of examples of the ‘aberrant structures’ mentioned in this sentence?

We narrowed the concept by specifying that these ‘aberrant structures’ derive from replication stress, whereas the DNA breaks generated during meiosis are quite physiological. We do not think that naming some of these structures is beneficial for the reader. If the reader is not an expert, listing fork collapse or R-loops, or the replication stress due to G-quadruplex DNA or common fragile sites will generate confusion and a thorough description is beyond the focus of this review. We suggest an excellent review as a reference for further reading on this topic for readers who are interested in it.

- line 38. It is mentioned that Mre11 exerts exo- or endonuclease functions: it might be relevant to precise that this is for short-range resection, in order to contrast with the recruitment of Exo1 or Dna2 exonucleases mentioned in line 34.

Thank you for the useful suggestion. This is now mentioned in the text, as well as its requirement for the release of blocked DNA ends.

- lines 39-42. Is there any hypothesis about why the MRN/X complex can not activate Tel1/ATM when bound to nucleosomal DNA?

Unfortunately, the mechanism of both binding and activation of Tel1/ATM is still unclear. We added a novel 4.6 section where we discuss this topic further, since we believe is one of the still unresolved issues in the DNA damage response field. We thank the reviewer for addressing this point here and in the following remarks.

- lines 43-45. A few more details about the Tel1/ATM mechanisms would help the reader to understand the importance of this player in the DSB repair mechanisms, and further underline the importance of MRN/X in the whole process.

We agree. We focused better on the Tel1/ATM activation by MRN/X in the novel section 4.6. However, we decided to mention the role of Tel1/ATM in controlling resection and the DDR only in the introduction. As you previously mentioned, this review is indeed focused on the dynamics properties of the Mre11/Rad50 complexes. A full description of Tel1/ATM role in DNA damage checkpoint would require too much space. We suggest additional reading on this topic as a reference and mention the relationship to human health in the Conclusions section.

- line 48. The ‘MR’ acronyme has not been defined before. What is the difference between MR and MRN/X ?

We substituted with Mre11/Rad50 when appropriate. We defined the abbreviations in the text when necessary.

- Figure 1. I am not sure about the background color choice here. It makes it difficult to read some of the labels. Maybe it could be changed into simple squares? It is not clear if we see in the pictures the case of eukaryotes or prokaryotes, it would be beneficial to harmonize this and state it clearly in the caption. Sae2/CtIP mentioned in the caption are not mentioned in the main text, nor on the figure: either add something about it in the text/on the figure or remove it. In the caption, line 59, it seems that Ku exerts an endo- and exonuclease activity: I think the sentence should be rephrased to avoid any misunderstanding. Overall I believe that the caption should be more concise and should describe more precisely the figure (ATP hydrolysis, resting vs cutting states…).

We thank the reviewer for his/her precious suggestions. We removed the background color. The model is general, but it is based on what was found in budding yeast, so we decided to mention this as the model organism. As a consequence, Sae2 requirement for Mre11 endonuclease activity is now included in the figure. The caption was rephrased to better describe what is in the figure. We are sorry that we were not able to be concise in this but we prefer the figure to be self-explicatory for the reader’s benefit.

- Figure 2 should be moved to the end of section 2. The panel C should provide more details, especially related to the amino acids mentioned in the main text (lines 90-103). Labels in panel C should specify which conformation is prone to DNA scanning and which is prone to DNA binding. Walker B in blue is not visible in the right monomer, please change the color. The conserved residues of the signature region should be discussed in the main text, which is not the case.

Figure 2 was moved. The Cys residues in the Zn-hook were labelled. The information suggested was added. Unfortunately, we had to label many regions in Rad50 and ran out of colors. We used a very dark blue. The panel B is now more descriptive of the signature position with respect to the nucleotide.

The conservation of the signature is well known and described in several works. We mentioned the consensus in the text.

- lines 84-88. More details about the function of the two states (ATP before/after hydrolysis) described here would be appreciated. It is not clear how ATP hydrolysis regulated rad50 dimerization and further interaction with Mre11.

We mentioned here that the conformation of Rad50 depends on the bound nucleotide but we prefer to talk more specifically about the Mre11 interaction with Rad50 dimer in the section 4.

- line 97. Phosphorylation of which residue ?

Thr690. We mentioned it in the text.

- line 100. “The human RAD50 coiled coil is characterized by two interruptions at 70 and 110 aa from the hook”. This is not clear; displaying these residues in Figure 2C would help the reader.

Unfortunately, a structure of the coiled-coil in human Rad50 is available only for the last 100 aa, due to its extension and extreme flexibility, indeed. Although the positions were described according to AFM analysis, in the structure the actual distance is different (45 aa from the hook for the first interruption) so we decided to remove the affirmation. Anyway, we show the structure since the coiled-coil interruption is quite peculiar, affecting one of the two arms but bending the other one. The message of this paragraph is that the CC region is interrupted at more than one point, conferring flexibility to this region.

- lines 101-103. How is this flexibility important for the protein function?

We moved to this section the sentence ‘Interestingly, a mutant affected by the deletion of both coiled-coil arms but still retaining the zinc-hook region was unable to create DNA loops or bridges between two DNA ends, suggesting that the extension or the flexibility of the coiled-coil regions is also involved in establishing these contacts’

- line 108. It is not clear how much conserved is the arginine mentioned here? ‘revealed an allosteric network clusters residues in the α1-β4 loop, in the vicinity of ATP binding, to residues at the base of the coiled-coil, together with Mre11 interacting domains and Rad50 dimerization regions.’ this sentence is not clear, please rephrase. Clearly depicting the α1-β4 loop in Figure 2 would help to understand the allosteric mechanism mentioned here.

We included the α1-β4 loop in Figure 2. We mention the Arg805 in P. furiosus, which is corresponding to R1217 in S. cerevisiae and R1214 in human Rad50 (as we mentioned in the text). It is also conserved in M. jannaschii, C. thermophilum, but not in E. coli.

- lines 113-118. Please separate this long sentence in two.

Yes, thank you for the suggestion.

- line 124. Tel1 (ATM) is mentioned previously as Tel1/ATM or ATM/Tel1, only as Tel1 afterwards, please harmonize.

 As for MRN/X, if we only mention Tel1 we now specify that we are talking from yeast experimental data, and viceversa for ATM and mammalian.

- line 125. How can Rad50 impact the NHEJ pathway ? It is not mentioned that it is involved also in this pathway in the text.

Indirectly, due to its role in DNA tethering. We substituted with DNA tethering, since it is the direct function exerted by Rad50 CC.

- Section 2.2 title does not translates what is discussed in this section (Tel1/ATM activation and additional functions of the coiled-coil region of Rad50). Please modify to make it clearer.

Sorry for the mistake. We substituted the title and reorganized the paragraph with the description of research regarding the coiled-coil and Zn-hook. We hope the manuscript will be easier to follow.

- Figure 3 should be move to the end of section 3. In caption of panel A, please cite the reference to the MD simulation study mentioned.

It was already mentioned but it was evidently not clear. We repeated it.

- line 141. The five motifs mentioned here are not visible in Figure 3A, adding an inset or coloring these motifs would help the reader here. Further description of these motifs would be beneficial.

The motifs were not included in the figure but in order to mark the active site we encircled it and made the Mn2+ ions more evident. In order to describe these motifs we would need a MSA, but it is already present in several papers in literature, so we decided to only show them in the figure and introduce a very brief description here of what is present in this region.

- line 143. ‘The central region of Mre11 contains the capping domain, which is involved in making contact with the 3’ DNA end.’ this region does not look central in Figure 2A, why ? Does ‘central’ relates to the primary sequence of the protein here?

It refers to the primary sequence. We specified ‘sequence’ after Mre11.

- line 144. The RBD should also appear on Figure 3 to help the reader understanding the overall structure of the complex.

The RBD is not resolved in eukaryotes due to the high flexibility, and never if not in association with Rad50. We indicate to see the following section, since the RBD is well resolved only in the E. coli structure of the Mre11/Rad50 homologs complex.

- line 148. Please cite the MD simulations work mentioned here.

We mentioned it.

- line 150. Reference to Figure 3B is not helping as the mutation is not depicted in Figure3. The impact of R10T onto the dynamics of the system should be described more into details here to help understanding its overall effect on the Mre11 function.

Sorry for the misunderstanding. The Figure shows the physiological rotation in the wild-type protein. We moved the reference to the Figure accordingly. The dynamics of the R10T mutant are not described here, we mention it since it is the demonstration that the capping rotation has a function in DNA unwinding and Exo1 recruitment.

- line 156. So R10T activate the recruitment of Exo1 to the unwinded DNA ? I am not sure to fully understand this paragraph. A more detailed description of the reported observations would be beneficial here.

We rephrased, reporting some of the experimental data regarding the mutant R10T. The rotation we describe is physiological in the wild-type, and the R10T phenotype revealed that it is accessory in allowing Exo1 attack.

- Figure 4 to the end of section 4.3. Please cite the cryoEM publications associated with the structures displayed. It is not clear in which state are Rad50 coiled coils discussed previously and how it relates to the overall complex formation and DNA binding in terms of structure. There are missing residues in the experimental structures used for this figure, it should be mentioned. The ‘endonuclease’ and ‘exonuclease’ states are not discussed in the main text, but rather the ‘DNA end-associated’ and ‘distal DNA-associated’ complexes: please harmonize this.

We added all the information suggested. We focused better on the coiled coil arrangement in the text at several points, but it is actually not entirely clear when and how the coiled coil clamps to a rod. In all the structures, ATP-bound MRN/X displays oval-shaped coiled coils, but it rarely is bound to DNA. When DNA is present, hydrolysis happens and the structure has rod-coiled coils and the MRN/X structure is bound to the DNA, but it was recently affirmed that DNA binding elicits coiled coil opening AND ATP hydrolysis. We discuss this in the text in section 4.4.

- line 171. ‘in protein-protein interaction’ with whom ?

With Rad50 itself. We rephrased.

- lines 172-179. Any clue about where the binding site for Tel1?

We explained this point in the novel 4.6 section.

- line 191-192. What is the mentioned drift of the Mer11 subunit related to in terms of function? What role has it got in the overall mechanism?

We explained that the drift is required for the rotation of Mre11 dimer along Rad50 dimer required to reach the cutting state conformation.

- line 197. Any possibility to illustrate this in Figure 4 ?

We changed figure 4 since it was impossible to visualize the fastener in that orientation of the structure. In order to better appreciate the cutting state conformation, which is quite different than the resting state, we decided to show two different orientations of the most recent structure representing the endonuclease competent state. The fastener is now highlighted in Figure 4 panels B, C and D.

- line 199. Sae2 is not mentioned previously (only in a caption). It would be good to add something about it in the introduction.

We mentioned Sae2 role in endonuclease activity of MRN/X in the introduction.

- l213. How does Sae2 negatively regulate the DDR if it favors MRN/X endonuclease activity and Dna2 exonuclease recruitment ?

We added the role of Sae2 in counteracting Rad53 phosphorylation by Rad9.

- l215. Mre11 fastener loop is not well described in section 3. It should be, as it is mentioned here.

We specified the extension of the loop, which is now shown in Figure 4.

- l217. Sae2 interaction with whom?

Rad50. We specified it.

- l220 and after. MRX = MRN/X?

Rif2 role was clarified in yeast. So we are talking of Tel1 and MRX. We mentioned it in the text.

- line 238. Define SMC.

We added the definition and some more information.

- line 244. CtMR relation with the MRN/X complex should be introduced first here.

We explained better what we intended with CtMR.

- lines 245-246. How can DNA binding be DNA-independent?

Sorry for mistyping. It was supposed to be ‘ATP-independent’.

- l257. Add a reference.

Added, thank you for suggesting it.

- In Figure 5, the length of the coiled-coil is much longer than what can be observed in the other figures. If it is a matter of missing residues in the previous figures, it should be mentioned. How where these models built ? Captions 5C and D, add references.

We added the note on the missing residues in the captions of the previous figures. As mentioned above, the CC is very long, especially in eukaryotes. This figure shows cartoons, not structural models. The length of the coiled coil represented is in scale with the globular head dimension and the DNA molecule for a eukaryotic complex. The Figure 5D depicts the situation described for CtMR, and is based on a structure of ATP-bound CtMR. We added this information in the captions along with references.

- lines 294-298. This part, about interaction with Tel1, should be moved to the previous paragraph about this point.

Now this topic is discussed in detail in 4.6, so we direct to the following section.

- Figure 6 B and C are identical, only rotated. Please modify this figure, it is really not clear. This figure is very hard to understand with respect to the main text.

We modified the figure. The starting point and the final point are identical, for sure: these are the conformations that are available in CryoEM structures. The figure is meant to illustrate how the complex can move from one conformation to the other, and which are the topological problems arisen by each path. We hope that it is more straightforward in the revised version.

- Section 4.5 should be located after section 4.2.

It would be reasonable, but it is difficult to describe the problem of the conformational change from resting to cutting state without the previous discussion of the topology of the DNA binding by MRN/X complexes. Moreover, we think that the 4.5 section describes an open issue, as well as 4.6 section, so we feel that it is reasonable that they are at the end. We moved all the discussion of the topological path from resting to cutting state in the 4.5 section.

- lines 324-325. DNA repair has actually been studied since a very long time for therapeutic purposes. This paragraph should be removed.

We removed it.

- line 329-330. Why mentioning NPC here and nowhere else in the manuscript?

We removed it.

- The Conclusion section must be re-written. It does not summarize the topic reviewed nor gives any perspective about it.

Thank you for your comment. We rewrote the conclusion section. We feel that these novel roles are interesting and reveal how the complexity of MRN/X functions is not yet fully understood, so we would like to maintain a brief mention to them here.

Reviewer 2 Report

This paper is a review of the Mre11/Rad50 2 complex and will be of great benefit to non-specialists. It is a reader-friendly paper and many of the inserted figures are appropriate. I would consider it an acceptable paper if the following points were corrected.

1) Figure 2B is confusing; in addition to Figure 2B, a more detailed panel of correlation effects (e.g. hydrogen bonding) by chemical structure using e.g. Chemdraw should be added to Figure 2.

2) The meaning of the arrows in Figure 3 is not clear. It should be changed to a figure that is easier to understand and more intuitive for the reader.

3) The meaning of the arrow in Figure 6A is not clear. It should be changed to a figure that is easier to understand and more intuitive for the reader.

Suggestion: Although it is not related to the peer review of this paper, the authors are researchers who are capable of examining the current status in detail and writing a review article, and I would like to see a review article on Ku proteins that are in the prestage of the Mre11/Rad50 complex.

Author Response

- This paper is a review of the Mre11/Rad50 2 complex and will be of great benefit to non-specialists. It is a reader-friendly paper and many of the inserted figures are appropriate. I would consider it an acceptable paper if the following points were corrected.

We thank the reviewer for his/her appreciation.

1) Figure 2B is confusing; in addition to Figure 2B, a more detailed panel of -correlation effects (e.g. hydrogen bonding) by chemical structure using e.g. Chemdraw should be added to Figure 2.

A panel was prepared with the detail of the residues described in the text.

2) The meaning of the arrows in Figure 3 is not clear. It should be changed to a figure that is easier to understand and more intuitive for the reader.

We annotated the figure better.

3) The meaning of the arrow in Figure 6A is not clear. It should be changed to a figure that is easier to understand and more intuitive for the reader.

We changed the figure to better illustrate the two topological paths we describe in the text.

Round 2

Reviewer 1 Report

The manuscript has been modified as suggested. I believe it is of high quality and ready for publication.

One very minor thing: some of the references contain the doi, some don't. It should be harmonized according to IJMS criteria.